# SCOREFLOW: BRIDGING SCORE AND NEURAL ODE FOR REVERSIBLE GENERATIVE MODELING

## ABSTRACT

Neural ordinary differential equations (ODEs) are commonly used in reversible generative models. However, training neural ODEs is computationally expensive for estimating the log-likelihood density and backpropagating through ODE solvers, leading to slow convergence and significant gradient estimation errors. This paper presents ScoreFlow, a novel generative model capable of reversible and controllable data transformations. Firstly, we formulate an ODE utilizing a score variant as the drift term to model transformations between two certain data distributions. Secondly, we suggest a path-constrained loss to reduce truncation errors, enhancing the model's capabilities in generating high-quality samples. Thirdly, ScoreFlow has the ability to employ a single model to achieve both conditional image generation and cross-class image translation tasks. The closed-form optimal solution for data transformation in ScoreFlow is theoretically proven, providing support for the model's efficient training. Furthermore, the effectiveness of our approach is empirically validated through image generation, translation, and interpolation experiments.

## 1 INTRODUCTION

Neural ODEs have been intensively used in generative tasks for modeling the transformation between different distributions (Chen et al., 2018; Grathwohl et al., 2018; Finlay et al., 2020). By solving the forward and backward initial value problems (IVPs) of the neural ODEs, these methods facilitate a continuous-in-time mapping process between samples and Gaussian noise. In particular, *Instantaneous Change of Variables* (Chen et al., 2018) offers a means to compute the log-likelihood of generating samples, enabling the training of neural ODEs through maximum likelihood estimation (MLE). Neural ODEs are fully reversible generative models, that offer advantages including exact likelihood computation, latent representation manipulation, and efficient sampling.

However, ODE-based generative models also encounter certain challenges. First, the likelihood estimation requires computing the divergence of the drift function, leading to computational expense and resulting in slow training speed. Second, the training procedure of neural ODEs is very complex, bringing about difficulties in converging and generating high-quality samples. Third, compared with score-based models Ho et al. (2020); Song et al. (2020c) and GANs (Goodfellow et al., 2014; Choi et al., 2020), current ODE-based methods cannot achieve some benchmark tasks such as conditional sample generation and image translation tasks, restricting their applications.

Additionally, score-based models, achieved through a stochastic differential equation (SDE) framework (Song et al., 2020c), use an iterative denoising procedure to achieve high quality data generation. The fundamental concept of score-based models is the score function, i.e. $\nabla_x \log p(x)$, intuitively interpreted as the direction of the gradient that maximizes the log-likelihood of the samples. Based on the properties of score and Bayes rules, it is possible to achieve controllable data generation through both classifier-guided (Dhariwal & Nichol, 2021) and classifier-free (Ho & Salimans, 2021) methods.

The drift function of neural ODEs represents the transformation dynamics between data distributions. It is essential for efficient generative modeling. However, the formulation of the drift function is not well explored in existing ODE-based methods. In this paper, we construct a novel reversible generative framework called *ScoreFlow*, which incorporates a score variant into neural ODEs as the drift function. We empower traditional neural ODEs with the training efficiency, reversibility,

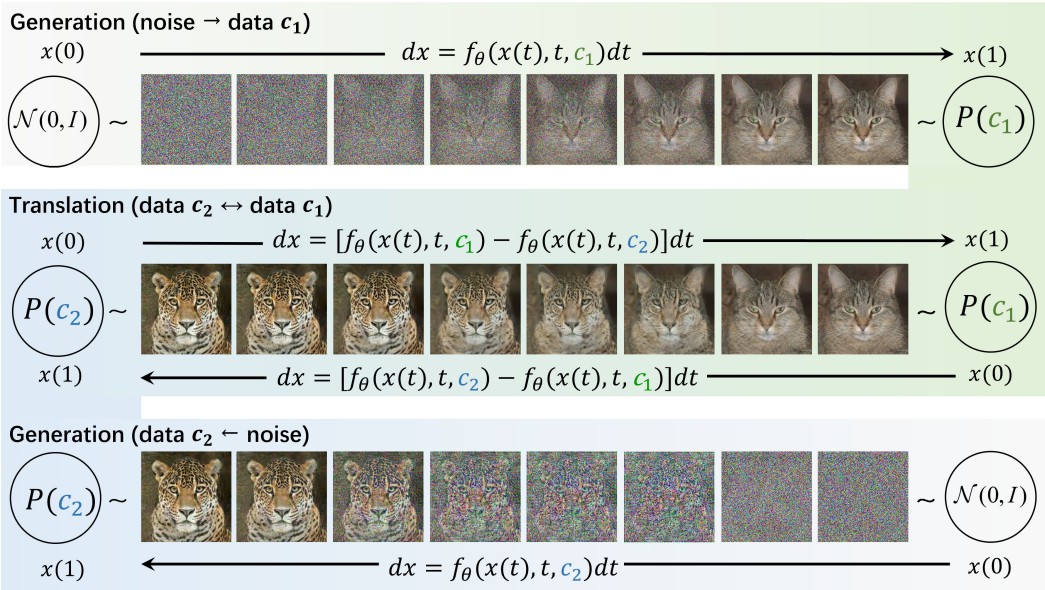

Figure 1: ScoreFlow can utilize one *unified* model $f_\theta$ to achieve both the conditional data generation and cross-class image translation. The drift of ODE is represented by a neural network for estimating a score variant $f_\theta \propto \nabla_x \log \frac{p_{data}(x(t))}{\mathcal{N}(0,I)}$. This method can effortlessly transform data between different domains by subtracting the respective drift to create the transformation ODE.

and versatility for high-quality sample generation. As depicted in Figure 1, via a single conditional model, ScoreFlow achieves both conditional sample generation and reversible image translation between selected classes. Our contributions can be summarized as follows:

- We introduce ScoreFlow, a novel framework for reversible generative modeling that bridges the gap between score and neural ODE. We derive a theoretical solution of ScoreFlow that provides a simple MSE method for training the neural ODE.

- To achieve high quality data generation, we propose a novel path-constrained loss to minimize truncation error in solving ODEs. Experimental results on CIFAR10 show that our method achieves higher metrics (FID of 2.29 & IS of 9.96) compared to other ODE-based and SDE-based algorithms.

- We achieve class-conditional sample generation and cross-class image translation utilizing a single model. The image translation can be performed between any two selected classes in the dataset with no additional effort. The experimental results of image translation and image interpolation on $256 \times 256$ datasets demonstrate the reversibility and versatility of ScoreFlow.

## 2 BACKGROUND

### 2.1 NEURAL ODEs & CONTINUOUS FLOW

Continuous Normalizing Flows (CNFs) (Chen et al., 2018; Grathwohl et al., 2018) are a type of generative models that employ neural ODEs to model the transformation between distributions. Let $z(0) \in \mathbb{R}^d$ denote the sample from a given dataset, and $z(T) \in \mathbb{R}^d$ denote the latent variable sampling from some tractable distribution. CNFs aim to construct a continuous sequence $\{z(t)\}_{t=0}^T$ mapping between the sample and the latent variable. In the forward process, CNFs represent the continuous-in-time transformation from sample to latent variable by an ODE:

$$dz(t) = f_\theta(z(t), t)dt, \tag{1}$$

where the drift $f_\theta(\cdot, t) : \mathbb{R}^d \to \mathbb{R}^d$ is a $d$-dimensional vector function parameterized by $\theta$. Given $z(0)$ as the initial input, $z(T)$ can be computed by solving the following initial value problem (IVP):

$$z(T) = z(0) + \int_0^T f_\theta(z(t), t)dt, \tag{2}$$

which can be solved via a numerical ODE solver like *Runge–Kutta*. In the reverse process, CNFs sample a prior variable $z(T)$ and generate data by solving the reverse IVP from $t = T$ to $t = 0$. *Instantaneous Change of Variables* (Chen et al., 2018) offers an approach for computing the log-likelihood of generated data. Thus, CNFs can be trained utilizing maximum likelihood estimation.

Training the drift function $f_\theta$ requires propagating gradients through the ODE solver, which is very computationally expensive for large-scale generative tasks. *Rectified Flow* (RectFlow) (Liu et al., 2023) presents a simple but effective scheme that learns the ODE following the most straight velocity field as possible. The objective of RectFlow is to solve the following least squares regression problem:

$$\begin{aligned} \min_\theta \quad & \mathbb{E}_{(t,z(t))}[|| (z(0) - z(T)) - f_\theta(z(t), t)||^2], \\ s.t. \quad & z(t) = t \cdot z(T) + (1 - t) \cdot z(0), t \in [0, T], \end{aligned} \tag{3}$$

where we use the same denotations as in (1). This approach reduces the transportation cost and the computational demand in training phase. Furthermore, applying RectFlow recursively helps to straighten the flow path, which can reduce numerical errors in data generation and decrease the number of sampling steps (Liu et al., 2023).

## 2.2 SCORE-BASED MODELS

Score-based models typically include a forward diffusion process and a backward sampling process. The forward process constructs a sequence $\{z(t)\}_{t=0}^T$ for gradually corrupting the data to Gaussian noise, following the SDE:

$$dz(t) = f(z(t), t)dt + g(t)dw, \tag{4}$$

where $f(\cdot, t) : \mathbb{R}^d \to \mathbb{R}^d$ is a predefined $d$-dimensional vector function called the drift coefficient, $g(\cdot) : \mathbb{R} \to \mathbb{R}$ is a predefined scalar function called the diffusion coefficient, $w$ denotes the Wiener process, and $t \in [0, T]$ is the timestep index. In the backward process, score-based models first samples $z(T)$ from the latent distribution and then apply recursively denoising to generate sample $z(0)$, following the reverse-time SDE:

$$dz(t) = [f(z(t), t) - g^2(t)\nabla_{z(t)} \log p_t(z(t))]dt + g(t)dw, \tag{5}$$

where $\nabla_{z(t)} \log p_t(z(t))$ is called the *score* function. Score-based models train a neural network $s_\theta(z(t), t)$ to estimate the score function via minimizing the following loss:

$$\theta^* = \arg\min_\theta \mathbb{E}_{t,z(t)}[\lambda(t)||s_\theta(z(t), t) - \nabla_{z(t)} \log p_t(z(t))||^2], \tag{6}$$

where $\lambda(t)$ is a positive weighting function.

## 3 METHOD

### 3.1 OVERVIEW

Our motivation is to explore an effective formulation of the drift $f_\theta$ in neural ODE (1), with the objective of providing ODE-based models with high-quality sample generation capabilities and a more simplified training method. Inspired by the drift coefficient in reverse-time SDE (5) of score-based models, we intend to formalize the drift of ODE to guide the direction that maximizes the log-likelihood of samples. Given two data distributions $x_1 \sim p_1(x_1)$ and $x_2 \sim p_2(x_2)$, to achieve the transformation between them, we employ the following formulation of ODE with a score variant as the drift:

$$\frac{dx(t)}{dt} = \dot{\sigma}_t \sigma_t \nabla_x \log \frac{p_2(x(t))}{p_1(x(t))}, \tag{7}$$

where $\sigma_t$ is a deviation function satisfying certain conditions (for details see subsection 3.2). The drift function this ODE (7) can be viewed as a scaled difference between two score functions, hence

we name this approach ScoreFlow. However, as the marginal distribution $p_1$ and $p_2$ are unknown, we cannot employ this ODE directly for generative modeling. Consequently, we instead train a neural network $f_\theta(x(t), t)$ to estimate the drift in Equation (7). Actually, we derive a simple closed-form solution to this drift as follows:

$$\dot{\sigma}_t \sigma_t \nabla_x \log \frac{p_2(x(t))}{p_1(x(t))} = \frac{\dot{\sigma}_t}{\sigma_t}(x_2 - x_1), \tag{8}$$

which can be used for training the network. Inspired by the training method (3) in RectFlow, we develop a path-constrained loss to learn a "straight" vector field:

$$\begin{aligned}
\min_\theta \quad & \mathbb{E}_{t,x(t)}\left[\lambda(t)\|x(t) - \bar{x}(t)\|^2\right], \\
s.t. \quad & dx(t)/dt = f_\theta(x(t), t), \quad x(0) = x_1 \\
& \bar{x}(t) = (1-t)x_1 + tx_2, \quad t \sim Uniform[0, 1]
\end{aligned} \tag{9}$$

where $\lambda(t)$ is a positive coefficient function, $x(t)$ is the intermediate state estimated by the ODE, and $\bar{x}(t)$ is the target intermediate state computed by a linear interpolation between $x_1$ and $x_2$. Furthermore, for achieving class-conditional data generation, we can straightforwardly embed the class label $c$ to train a conditional neural network $f_\theta(x(t), t, c)$ (for details see subsection 3.2).

Figure 1 illustrates a specific generation-translation task achieved by our method. After training, ScoreFlow is capable of using a single model to achieve both the class-conditional generation and cross-class image translation. In the conditional generation process, we initially sample a latent variable from $\mathcal{N}(0, I)$ as the initial state, and then solve the following ODE from $t = 0$ to $t = 1$:

$$dx = f_\theta(x(t), t, c)dt, \tag{10}$$

where $c$ denotes the corresponding class label. In the image translation task, we can reuse this network $f_\theta$ along with the class labels to create the transformation ODE between different domains. Specifically, translating data from $c_1$ to $c_2$ requires to solve the following ODE from $t = 0$ to $t = 1$, with a $c_1$-class sample as the initial state:

$$dx = [f_\theta(x(t), t, c_2) - f_\theta(x(t), t, c_1)]dt. \tag{11}$$

The reverse process employs the same approach.

### 3.2 BRIDGING ODE WITH SCORE FUNCTION

In this subsection, we detail how to bridge the gap between neural ODEs and the score function to improve the performance of generative modeling based on ODEs. The score function establishes a connection between the generative processes and the direction of log-density. This concept forms the cornerstone of score-based generative models (Hyvärinen & Dayan, 2005; Vincent, 2011; Song & Ermon, 2019), that achieve high-quality sample generation. Intuitively, the score function provides a direction that guides the intermediate state to move in a way that leads to higher log-density. Based on this property, it is possible to achieve conditional generation through both classifier-guidance (Dhariwal & Nichol, 2021) and classifier-free (Ho & Salimans, 2021) approaches. We first establish the relationship between the ODE and the score function, and derive the formulation of ScoreFlow. We then present path-constrained loss for training ScoreFlow. Building upon this, we propose a novel method for conditional data generation that relies entirely on ODE.

**Lemma 3.1.** *Given an ODE $\frac{dx(t)}{dt} = g(x(t), t)$ with a Lipschitz continuous drift $g(x(t), t)$, the probability density $p_t(x)$ satisfies the continuity equation (Pedlosky, 2013):*

$$\frac{\partial p_t(x(t))}{\partial t} + \nabla \cdot p_t(x(t))g(x(t), t) = 0 \tag{12}$$

To establish a connection between this ODE and the score, we assume the drift function has the formulation of $g(x(t), t) = -D_t \nabla_x \log p(x(t))$, where $D_t$ is a positive diffusion coefficient. Substitute this into (12), we can obtain the following equation:

$$\frac{\partial p(x(t))}{\partial t} = D_t \nabla^2 p(x(t)), \tag{13}$$

where $\nabla^2$ denotes Laplace operator. It is worth noting that (13) corresponds to the Fokker–Planck equation (Risken & Risken, 1996) with a zero drift, which can be solved analytically. Therefore, we can obtain the exact probability density of the intermediate state $x(t)$ at arbitrary time $t$, bringing our great attention to this particular form of ODE.

---

**Algorithm 1** Class-Conditional ScoreFlow

---

**Input**: parameterized drift function $f_\theta(\cdot)$, images $x \sim p_{data}$, corresponding class labels $c$
**Training**
   **repeat**
      $z \leftarrow$ sample from $\mathcal{N}(0, I)$
      $t \leftarrow$ sample from Uniform[0,1]
      $\bar{x}(t) \leftarrow (1-t)z + tx$
      $x(t) \leftarrow z + t f_\theta(\bar{x}(t), t, c)$
      $L^p(\theta) \leftarrow mean(\frac{1}{t}||x(t) - \bar{x}(t)||^2)$
      apply gradient descent step on $\nabla_\theta L^p(\theta)$
   **until** converged
**Sampling for Image Generation**
   $z \leftarrow$ sample from $\mathcal{N}(0, I)$
   generate data $x^c$ solving $x^c = z + \int_0^1 f_\theta(z(t), t, c)dt$ via some ODE solver
**Sampling for Image Translation**
   $c_1 \leftarrow$ label of source domain, $c_2 \leftarrow$ label of target domain, $x^{c_1} \leftarrow$ images of class $c_1$
   generate data $x^{c_2}$ solving $x^{c_2} = x^{c_1} + \int_0^1 [f_\theta(x(t), t, c_2) - f_\theta(x(t), t, c_1)]dt$ via some ODE solver

---

**Theorem 3.1.** *Suppose the evolving of $x(t) \in \mathbb{R}^d$ satisfies an ODE $\frac{dx(t)}{dt} = -D_t \nabla_x \log p(x(t))$, the conditional probability density of $x(t)$ given $x(0)$ is:*

$$p(x(t)|x(0)) = \mathcal{N}(x(t); x(0), \sigma_t^2 I) = \frac{1}{(2\pi\sigma_t^2)^{d/2}} \exp(-\frac{||x(t) - x(0)||^2}{2\sigma_t^2}). \quad (14)$$

*Given two data distributions $x_1 \sim p_1(x_1)$ and $x_2 \sim p_2(x_2)$, based on Equation (14), the ScoreFlow mapping data from $x_1$ to $x_2$ can be derived as:*

$$\frac{dx(t)}{dt} = \dot{\sigma}_t \sigma_t \nabla_x \log \frac{p_2(x(t))}{p_1(x(t))} = \frac{\dot{\sigma}_t}{\sigma_t}(x_2 - x_1), \quad (15)$$

*where $\sigma_t \geq 0$ denotes a monotonic increasing function with $\sigma_t \gg 1$ as $t \to \infty$, $d$ denotes the number of dimensions.*

See the supplementary material for the proof.

**Discussion** Theorem 3.1 presents the analytical formulation of ScoreFlow mapping the data from $p(x_1)$ to $p(x_2)$. In practical generative process, $\frac{\dot{\sigma}_t}{\sigma_t}(x_2 - x_1)$ is not a causal item because we cannot obtain $x_2$ before it is generated. Therefore, we approximate the drift function in (15) using a parameterized neural network $f_\theta(x(t), t)$. To learn the parameters $\theta$, we construct a simple MSE loss to reduce the $L^2$-norm of the difference between $f_\theta$ and $\frac{\dot{\sigma}_t}{\sigma_t}(x_2 - x_1)$ as follows:

$$\min_\theta \mathbb{E}_{t,x(t)} \left[ ||f_\theta(x(t), t) - \frac{\dot{\sigma}_t}{\sigma_t}(x_2 - x_1)||^2 \right]. \quad (16)$$

After training $f_\theta$, we solve the following IVP via some ODE solver to transform data from $x_1$ to $x_2$:

$$x_2 = x_1 + \int_0^T f_\theta(x(t), t)dt. \quad (17)$$

Inspired by RectFlow, we aim to learn a drift function $f_\theta$ that is as straight as possible Liu et al. (2023). Under this objective, the coefficient factor is a constant $\dot{\sigma}_t/\sigma_t = C$. If we constrain the integral interval as $[0, 1]$, i.e. $T = 1$, then $\dot{\sigma}_t/\sigma_t = 1$ and $\dot{\sigma}_t = \sigma_0 e^t$, which satisfies the conditions of Theorem 3.1. Then, the optimization problem becomes:

$$\min_\theta \quad \mathbb{E}_{t,x(t)} \left[ ||f_\theta(x(t), t) - (x_2 - x_1)||^2 \right], \\ s.t. \quad x(t) = (1-t)x_1 + tx_2, t \sim Uniform[0, 1], \quad (18)$$

It can be noticed that the loss (18) is the same as (3), indicating that RectFlow is a specific form of ScoreFlow.

**Path-Constrained Loss**   Based on the aforementioned discussion, it can be noticed that the naive MSE loss (18) is equivalent to optimize the $L^2$ error between a one-step Euler IVP solution and the ground truth, i.e. $||(x_1 + f_\theta(x(t), t)) - x_2||^2$. This loss solely accounts for the cumulative error of the ODE, neglecting the truncation error that arises in the intermediate paths. To tackle this concern, we introduce a path-constrained loss that incorporates truncation errors occurring during the forward process of the ODE. Since the intermediate state of the target ODE path can be painlessly obtained by a linear interpolation $x(t) = (1 - t)x_1 + tx_2$, we can compute the truncation error at arbitrary time $t$. Based on this, we present the following path-constrained loss function:

$$\min_\theta \quad \mathbb{E}_{t, x(t)} \left[ \lambda(t) || x(t) - \bar{x}(t) ||^2 \right],$$
$$s.t. \quad dx(t)/dt = f_\theta(x(t), t), \quad x(0) = x_1$$
$$\bar{x}(t) = (1 - t)x_1 + tx_2, \quad t \sim Uniform[0, 1] \tag{19}$$

To alleviate the computational load, we utilize a one-step approximation $x(t) \approx x_1 + tf_\theta(\bar{x}(t), t)$ to obtain the estimated intermediate state. We choose $\lambda(t) = 1/t$, for eliminating the influence of scaling $f_\theta$ by $t$ when backpropagating gradients through $x(t)$. Note that the path-constrained loss does not impose constraints on the form of the two distributions. In other words, $x_1$ can follow arbitrary data distribution, not just limited to Gaussian. This enables ScoreFlow to achieve image-to-image translation directly.

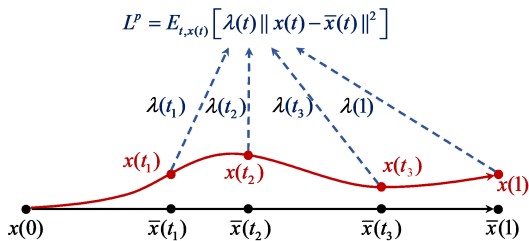

Figure 2: The path-constrained loss considers the truncation error that arises between the solution path and the target path while solving the ODE.

**Classifier-free Conditional Data Generation**   As shown in Equation (15), the drift $\dot\sigma_t \sigma_t \nabla_x \log \frac{p_2(x(t))}{p_1(x(t))}$ can be interpreted as a scaled difference between the score function of $p_2$ and $p_1$. We can directly introduce an additional control vector $c$ (which can be class label, text prompt, etc.) to the drift $f_\theta(x(t), t, c)$ for achieving conditional generation. Intuitively, the conditional drift can represent $\dot\sigma_t \sigma_t \nabla_x \log \frac{p_2(x(t)|c)}{p_1(x(t))}$, guiding the generation process to move to the region with high conditional log-likelihood. The algorithm is outlined by Algorithm 1.

### 3.3   Image-to-image Translation using a Unified Conditional Model

A significant advantage of the ScoreFlow is the capability of accomplishing cross-domain image translation through a single unified model. Suppose we have pre-trained a conditional ScoreFlow model $f_\theta$ on a multi-class image dataset. By controlling the input labels to the unified model, we can achieve the image translation between arbitrary two classes within the dataset.

**Theorem 3.2.** *Given a class-conditional ScoreFlow model $\frac{dx(t)}{dt} = f_\theta(x(t), t, c)$ which is pretrained for generating $x^{c_i} \sim p_{data}$ from some latent space $z \sim p_z$, where $c_i \in \{0, 1, 2, ..., N\}$ denotes the i-th class label. Then, the image translation from class $c_i$ to $c_j$ can be achieved by solving the following IVP:*

$$x^{c_j} = x^{c_i} + \int_0^1 [f_\theta(x(t), t, c_j) - f_\theta(x(t), t, c_i)]dt, \tag{20}$$

*where $x^{c_i}$ and $x^{c_j}$ denote images with different labels $c_i$ and $c_j$.*

*Proof.* The proof is straightforward and intuitive. Due to the invertibility of ODE, we have $z_0 = x^{c_1} + \int_1^0 f_\theta(x(t), t, c_1)dt$. Then we can obtain $x^{c_2} = x^{c_1} + \int_1^0 f_\theta(x(t), t, c_1)dt + \int_0^1 f_\theta(x(t), t, c_2)dt$. By exchanging the upper and lower limits of the definite integral, and utilizing the additivity property, we can get the final formulation $x^{c_2} = x^{c_1} + \int_0^1 [f_\theta(x(t), t, c_2) - f_\theta(x(t), t, c_1)]dt$. The proof is concluded. □

**Discussion**   Theorem 3.2 introduces a direct and effective method for achieving cross-domain image translation. More specifically, to accomplish image translation between multiple categories,

we can train a unified conditional model $f_\theta(x(t), t, c)$ using the collected multi-category images. Subsequently, we solve the IVP defined in (20) with the drift being the difference between the two specific class-conditional drift functions. The algorithm is outlined in Algorithm 1.

| Method | FID ↓ | IS ↑ | NFE ↓ |
|---|---|---|---|
| SDE-based model | | | |
| DDPM (Ho et al., 2020) | 3.21 | 9.46 | 1000 |
| (NCSN++) VE(Song et al., 2020c) | 2.38 | 9.83 | 2000 |
| VP (Song et al., 2020c) | 2.55 | 9.58 | 2000 |
| sub-VP (Song et al., 2020c) | 2.61 | 9.56 | 2000 |
| ODE-based model | | | |
| 1-RectFlow (Liu et al., 2023) | 2.58 | 9.60 | 127 |
| 2-RectFlow (Liu et al., 2023) | 3.36 | 9.24 | 110 |
| 3-RectFlow (Liu et al., 2023) | 3.96 | 9.01 | 104 |
| VP ODE (Song et al., 2020c) | 3.93 | 9.37 | 140 |
| sub-VP ODE (Song et al., 2020c) | 3.16 | 9.46 | 146 |
| (NCSN++) VE ODE (Song et al., 2020c) | 5.38 | 9.35 | 176 |
| **ScoreFlow** (unconditional) | 2.58 | 9.72 | 124 |
| **ScoreFlow** (conditional) | **2.29** | **9.96** | 127 |

Table 1: Results of ODE&SDE-based methods on CI-FAR10. Fréchet inception distance (FID) and Inception score (IS) assess the quality of images, Number of function evaluations (NFE) assesses the number of sampling steps.

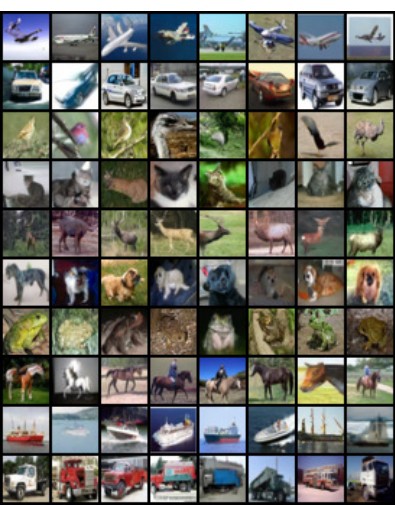

Figure 3: Generated samples by Score-Flow on CIFAR10.

## 4 EXPERIMENT

### 4.1 IMAGE GENERATION

In order to verify the effectiveness of the proposed model, we trained ScoreFlow on four datasets: CIFAR10(Krizhevsky & Hinton, 2009), AFHQ(Choi et al., 2020), MetFace(Karras et al., 2020), and CelebA(Karras et al., 2017). Images of the last three datasets are resized to $256 \times 256$.

**Setup** Following the algorithm 1, we set $z \sim \mathcal{N}(0, I)$ to be the latent variable and $x^c$ to be the images. We adopt DDPM++ (Song et al., 2020c) network architecture for representing the drift function $f_\theta$, which exhibits slight differences in terms of depth and the number of modules for different datasets. To introduce the conditional information to the model, we fuse the timestep embeddings and condition embeddings in the activations of each residual layer by

$$\tilde{a}_L = emb_c \odot a_L + emb_t, \tag{21}$$

where $\tilde{a}_L$ is the infused output of layer $L$, $\odot$ is the element-wise product, $emb_c$ and $emb_t$ are condition embedding and timestep embedding. This conditioning approach is inherited from Ho & Salimans (2021). We trained separate models for CIFAR10 and AFHQ datasets, using their respective category labels as conditions. For CelebA and MetFace, we combined them into a 2-class dataset and trained a unified model using the labels 0 and 1 as conditions. We also implemented the unconditional generation on CIFAR10 for comparison. For sampling, we utilize the *RK45* ODE solver from Scipy(Virtanen et al., 2020), and the tolerance is set to 1e-5.

**Result** Table 1 presents the FID and IS results of ScoreFlow in comparison to other ODE-based and SDE-based methods. On CIFAR10, conditional ScoreFlow yields the lowest FID (2.29) and highest IS (9.96) among all the methods, while the unconditional version also achieve the best performance among the ODE-based methods. It is worth noting that, compared to SDE-based methods, all the ODE-based methods have significant advantages in terms of sampling steps. The generated images are shown in Figure 3, 4 and 5.

### 4.2 IMAGE TRANSLATION & INTERPOLATION

**Setup** We directly utilize the models which are pre-trained in Section 4.1 for image translation and interpolation. For the unified model trained on CelebA and MetFace, we implement the translation

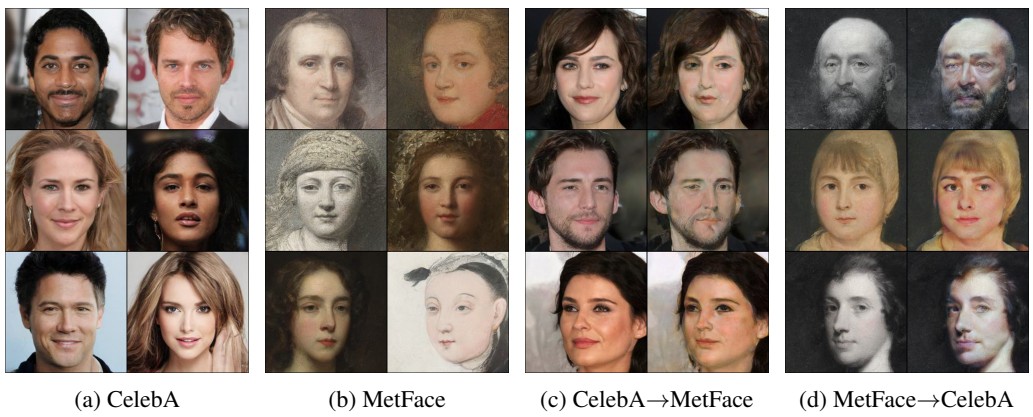

|              |              |                    |                    |
| :----------: | :----------: | :----------------: | :----------------: |
| (a) CelebA   | (b) MetFace  | (c) CelebA→MetFace | (d) MetFace→CelebA |

Figure 4: (a) and (b) The generated $256 \times 256$ samples by a unified conditional ScoreFlow $dx = f_\theta(x(t), t, c)dt$ with the labels setting to $c = 0$ and $c = 1$ respectively. (c) The image translation from CelebA to MetFace, via solving $dx = [f_\theta(x(t), t, c = 1) - f_\theta(x(t), t, c = 0)]dt$. (d) The reverse translation from MetFace to CelebA.

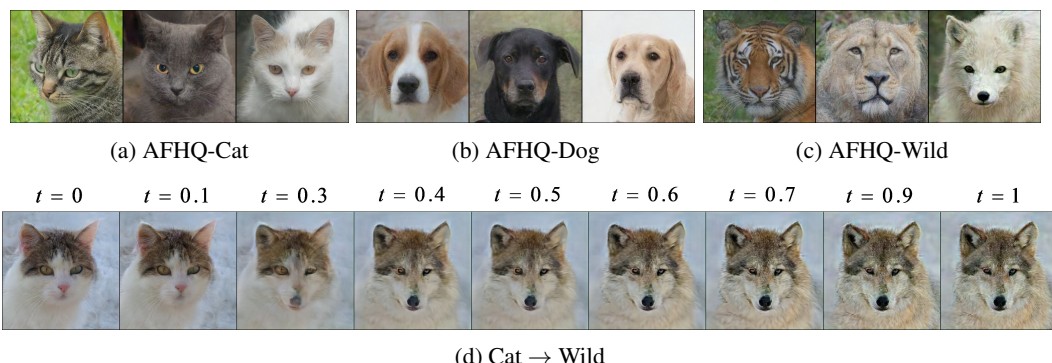

|              |              |                |
| :----------: | :----------: | :------------: |
| (a) AFHQ-Cat | (b) AFHQ-Dog | (c) AFHQ-Wild  |

(d) Cat → Wild

Figure 5: (a)-(c) The generated $256 \times 256$ samples by a conditional ScoreFlow $dx = f_\theta(x(t), t, c)dt$, where $c \in \{0, 1, 2\}$ is the label representing cat, dog and wild. (d) The transformation process from cat to wild, via solving $dx = [f_\theta(x(t), t, c = 2) - f_\theta(x(t), t, c = 0)]dt$

from CelebA to MetFace just by solving the ODE $dx = [f_\theta(x(t), t, c = 1) - f_\theta(x(t), t, c = 0)]dt$ from $t = 0$ to $t = 1$. The reverse translation is to solve the reverse ODE $dx = [f_\theta(x(t), t, c = 0) - f_\theta(x(t), t, c = 1)]dt$ from $t = 0$ to $t = 1$. The translation on AFHQ is the same way.

For image interpolation, we first sample two latent variables $z_0$ and $z_1$ from $\mathcal{N}(0, I)$, then conduct an interpolation of $z_\alpha = \sqrt{1 - \alpha}z_0 + \sqrt{\alpha}z_1$ with $\alpha \in [0, 1]$, and use these latents to generate images.

**Result** Figure 4c and Figure 4d show the translation between CelebA and MetFace. Figure 5d depicts the temporal progression of the transformation from a cat to a wild animal. Importantly, all image translations are effortless, as they can be achieved using the pre-trained generative model, without the necessity of training individual models for each specific transformation task. Figure 6 illustrates the variation process of image interpolation in the latent space.

## 5 RELATED WORK

**Neural ODEs & CNFs** Our work is based on neural ODEs that were first introduced by Chen et al. (2018). One significant application of neural ODE is its capability to construct CNFs. However, training CNFs needs to maximize the log-density of the generated samples, where the divergence computation is very expensive. FFJORD (Grathwohl et al., 2018) introduces Hutchinson's trace estimator to reduce the complexity of computing divergence, enabling scalable data generation for CNFs. Finlay et al. (2020) and Onken et al. (2021) focus on incorporating regularization to reduce

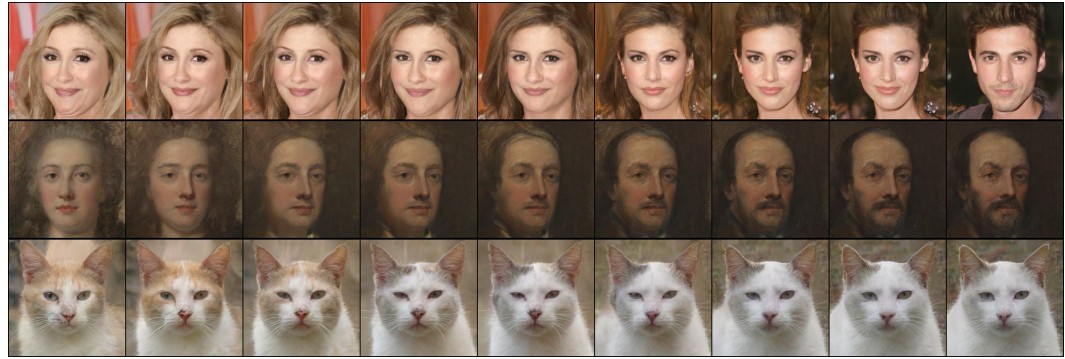

Figure 6: Image interpolation in latent space. The latent variable is set to $z_\alpha = \sqrt{1 - \alpha} z_0 + \sqrt{\alpha} z_1$ with $\alpha \in [0, 1]$, where $z_0$ and $z_1$ follow $\mathcal{N}(0, I)$.

the transport cost of the ODE and learn more straight transport path. Several studies concentrate on enhancing the termination time (output time) of the ODE, such as incorporating randomness (Ghosh et al., 2020) and employing adaptive optimization methods (Pang et al., 2022). In practical, the numerical errors in the process of solving ODE can affect the performance of the model, which has been partially resolved by checkpoint-based methods (Zhuang et al., 2020; Gholaminejad et al., 2019) or designing novel special integrator (Zhuang et al., 2021). Training Classical CNFs requires huge computational resources to compute the log-density and suffers from long training time. Rect-Flow (Liu et al., 2023) introduces a novel approach for training ODEs to learn a rectified vector field by employing a straightforward mean MSE loss function.

**Score-based & Diffusion Models**   Modern score-based and diffusion models exhibit a profound association with denoising score matching (Hyvärinen & Dayan, 2005; Vincent, 2011). The vanilla score matching approach suffers from expensive cost for computing the trace of Hessian. Sliced Score Matching (Song et al., 2020b) employs a projecting based method to address this limitation and achieves high dimensional density estimation. SMLD (Song & Ermon, 2019) presents multi-scale denoising approach, and employs *Noise Conditional Score Network* (NCSN) to learn the score function. When sampling, SMLD employs an annealed *Langevin dynamics* (Welling & Teh, 2011) to generate samples recursively using the learned score function. DDPM Sohl-Dickstein et al. (2015); Ho et al. (2020) implements a forward Markov noisy process and learns a reverse denoising network at arbitrary timesteps, achieving high quality image generation via recursively denoising sampling. Later, a comprehensive framework for score-based and diffusion models, based on SDEs, was developed by (Song et al., 2020c). The framework utilizes SDEs and score functions to describe both the forward and backward diffusion processes. Building upon these foundational works mentioned above, recent studies have made significant improvements, such as algorithmic efficiency enhancement (Karras et al., 2022; Nichol & Dhariwal, 2021; Song & Ermon, 2020; Song et al., 2023), maximum likelihood training (Lu et al., 2022a; Kim et al., 2022b; Song et al., 2021), sampling acceleration (Song et al., 2020a; Lu et al., 2022b; 2023), controllable generation (Dhariwal & Nichol, 2021; Ho & Salimans, 2021; Rombach et al., 2022; Kim et al., 2022a), and other downstream applications (Meng et al., 2021; Zhao et al., 2022; Ho et al., 2022; Harvey et al., 2022; Popov et al., 2021; Kong et al., 2020; Wyatt et al., 2022).

## 6   CONCLUSION

ScoreFlow effectively bridges denoising score-based models and neural ODEs, leading to reversible and conditional generative modeling. This approach leverages a variant of the score function to guide a neural ODE. By utilizing ODE's distinctive architecture, the model can acquire a theoretical solution that serves as an objective function for training the neural network. This objective guides the neural network to learn a "straight" vector field, resulting in an improved sampling speed. Furthermore, the incorporation of a path-constrained loss aids in reducing truncation errors and boosting model performance. ScoreFlow's versatility shines through its capacity for classifier-free conditional generation and seamless cross-class image translation within a unified framework. Empirical verification also attests to its efficacy in tasks such as image generation, translation and interpolation.

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
