# SUPPLEMENTARY MATERIAL FOR SUBMISSION #8904

## 1 PROOF OF THEOREM 3.1

**Lemma 3.1.** *Given an ODE $\frac{dx(t)}{dt} = g(x(t), t)$ with a Lipschitz continuous drift function $g(x(t), t)$, the probability density $p_t(x)$ satisfies the continuity equation:*

$$\frac{\partial p_t(x(t))}{\partial t} + \nabla \cdot p_t(x(t))g(x(t), t) = 0. \tag{1}$$

**Theorem 3.1.** *Suppose the evolving of $x(t) \in \mathbb{R}^d$ satisfies an ODE $\frac{dx(t)}{dt} = -D_t \nabla_x \log p(x(t))$, the conditional probability density of $x(t)$ given $x(0)$ is:*

$$p(x(t)|x(0)) = \mathcal{N}(x(t); x(0), \sigma_t^2 I) = \frac{1}{(2\pi\sigma_t^2)^{d/2}} \exp(-\frac{||x(t) - x(0)||^2}{2\sigma_t^2}). \tag{2}$$

*Given two data distributions $x_1 \sim p_1(x_1)$ and $x_2 \sim p_2(x_2)$, based on Eq.(2), the ScoreFlow mapping data from $x_1$ to $x_2$ can be derived as:*

$$\frac{dx(t)}{dt} = \dot{\sigma}_t \sigma_t \nabla_x \log \frac{p_2(x(t))}{p_1(x(t))} = \frac{\dot{\sigma}_t}{\sigma_t}(x_2 - x_1), \tag{3}$$

*where $\sigma_t \geq 0$ denotes a monotonic increasing function with $\sigma_t \gg 1$ as $t \to \infty$, $d$ denotes the number of dimensions.*

*Proof.* Without loss of generality, we first consider the case of dimension 1. According to Lemma 3.1, we first substitute the given ODE into Equation (1) and obtain:

$$\frac{\partial p(x(t))}{\partial t} = D_t \nabla^2 p(x(t)), \tag{4}$$

To solve this equation, we perform a spatial Fourier transform on $x$:

$$\mathcal{F}\left[\frac{\partial p(x(t))}{\partial t}\right] = D_t \mathcal{F}\left[\nabla^2 p(x(t))\right]$$
$$\implies \quad \frac{\partial}{\partial t}\mathcal{F}_t(\omega) = (-i\omega)^2 D_t \mathcal{F}_t(\omega). \tag{5}$$

By solving this ODE, we have:

$$\mathcal{F}_t(\omega) = \mathcal{F}_0(\omega)e^{-\omega^2 \int_0^t D_s ds}$$
$$\overset{\int_0^t D_s ds = \frac{1}{2}\sigma_t^2}{\implies} \quad \mathcal{F}_t(\omega) = \mathcal{F}_0(\omega)e^{-\frac{1}{2}\omega^2\sigma_t^2}. \tag{6}$$

Using the convolution theorem, we perform the inverse Fourier transform and obtain:

$$p(x(t)) = \int p(x_0)\frac{1}{(2\pi\sigma_t^2)^{1/2}} \exp(-\frac{|x(t) - x(0)|^2}{2\sigma_t^2})dx$$
$$\implies \quad p(x(t)|x(0)) = \frac{1}{(2\pi\sigma_t^2)^{1/2}} \exp(-\frac{||x(t) - x(0)||^2}{2\sigma_t^2}). \tag{7}$$

In multi-dimensional situations, we have:

$$p(x(t)|x(0)) = \frac{1}{(2\pi\sigma_t^2)^{d/2}} \exp(-\frac{||x(t) - x(0)||^2}{2\sigma_t^2})$$
$$= \mathcal{N}(x(t); x(0), \sigma_t^2 I), \tag{8}$$

and the ODE can be written as:

$$dx(t) = -\dot{\sigma}_t \sigma_t \nabla_x \log p(x(t)) dt. \tag{9}$$

Therefore, given $x(0) \in \mathbb{R}^d$ as the initial state of ODE (9), according to Equation (8), we can obtain $x(t)$ by computing:

$$x(t) = x(0) + \sigma_t \epsilon, \epsilon \sim \mathcal{N}(0, I). \tag{10}$$

If $\sigma_t \geq 0$ denotes a monotonic increasing function with $\sigma_t \gg 1$ as $t \to \infty$, when $t$ is sufficiently large, we have:

$$x(t) = \sigma_t \epsilon \tag{11}$$

Therefore, given $x_1 \sim p_1$ and $x_2 \sim p_2$ as the inital states of the ODE (9), when $T$ is sufficiently large, we have:

$$
\begin{aligned}
x_1(T) &= x_1 + \int_0^T -\dot{\sigma}_t \sigma_t \nabla_{x_1} \log p_1(x(t)) dt = \sigma_t \epsilon, \\
x_2(T) &= x_2 + \int_0^T -\dot{\sigma}_t \sigma_t \nabla_{x_2} \log p_2(x(t)) dt = \sigma_t \epsilon,
\end{aligned}
\tag{12}
$$

Therefore, we have:

$$
\begin{aligned}
x_2 &= x_1 + \int_0^T -\dot{\sigma}_t \sigma_t \nabla_{x_1} \log p_1(x(t)) dt - \int_0^T -\dot{\sigma}_t \sigma_t \nabla_{x_2} \log p_2(x(t)) dt \\
&= x_1 + \int_0^T \dot{\sigma}_t \sigma_t \nabla_x \log \frac{p_2(x(t))}{p_1(x(t))} dt \\
\implies \quad \frac{dx(t)}{dt} &= \dot{\sigma}_t \sigma_t \nabla_x \log \frac{p_2(x(t))}{p_1(x(t))}
\end{aligned}
\tag{13}
$$

Utilizing the Equation (8), we obtain:

$$
\begin{aligned}
\frac{dx(t)}{dt} &= \dot{\sigma}_t \sigma_t \nabla_x \log \frac{p_2(x(t))}{p_1(x(t))} \\
&= \dot{\sigma}_t \sigma_t \nabla_x \log \frac{\mathcal{N}(x(t); x_2, \sigma_t^2 I)}{\mathcal{N}(x(t); x_1, \sigma_t^2 I)} \\
&= \dot{\sigma}_t \sigma_t \frac{x_2 - x_1}{\sigma_t^2} = \frac{\dot{\sigma}_t}{\sigma_t}(x_2 - x_1)
\end{aligned}
\tag{14}
$$

This completes the proof. $\square$

## 2 MORE EXPERIMENTAL RESULTS

### 2.1 $256 \times 256$ IMAGES GENERATED BY A UNIFIED SCOREFLOW TRAINED ON CELEBA AND METFACE

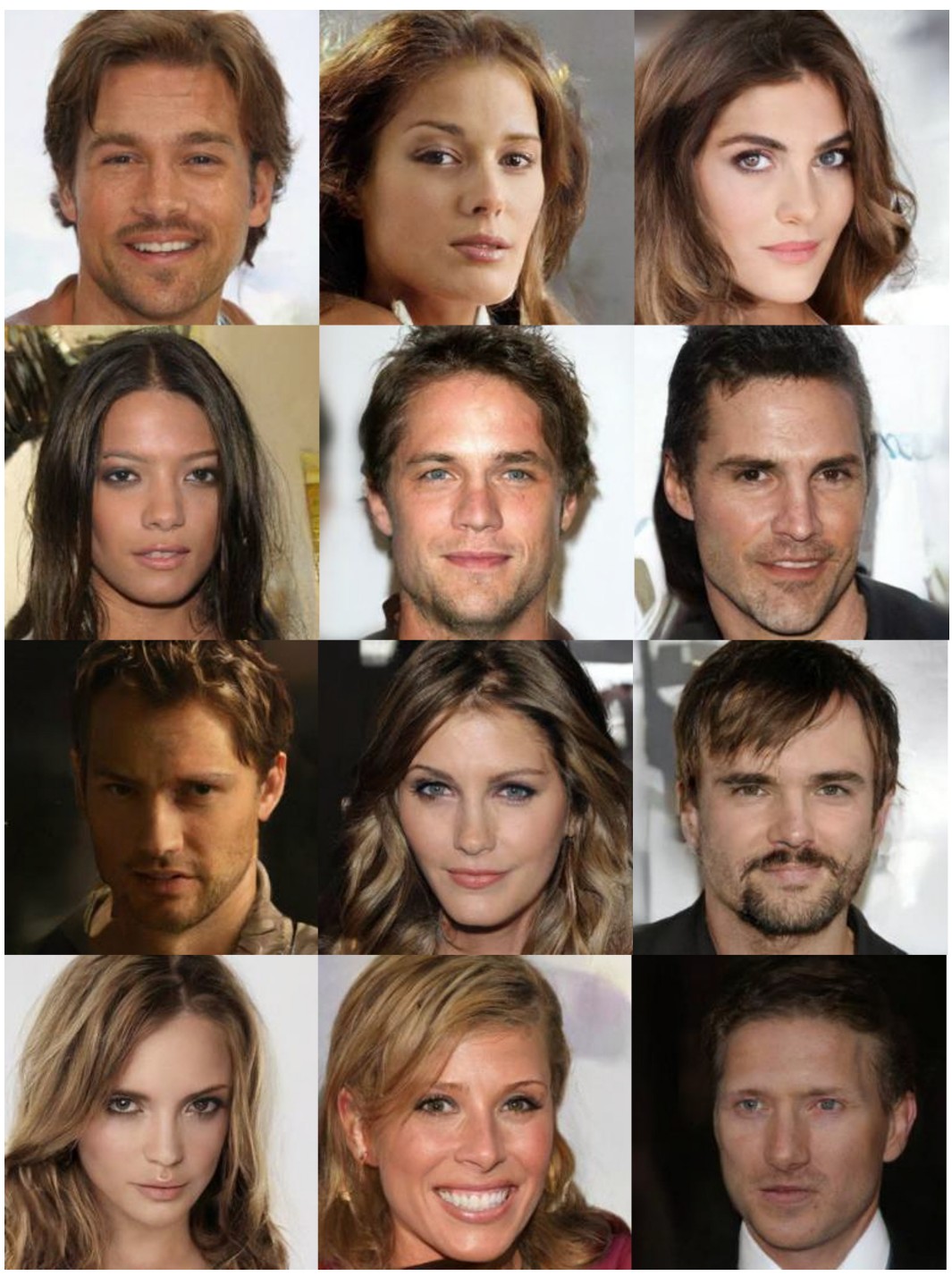

Figure 1: Generated Images on CelebA

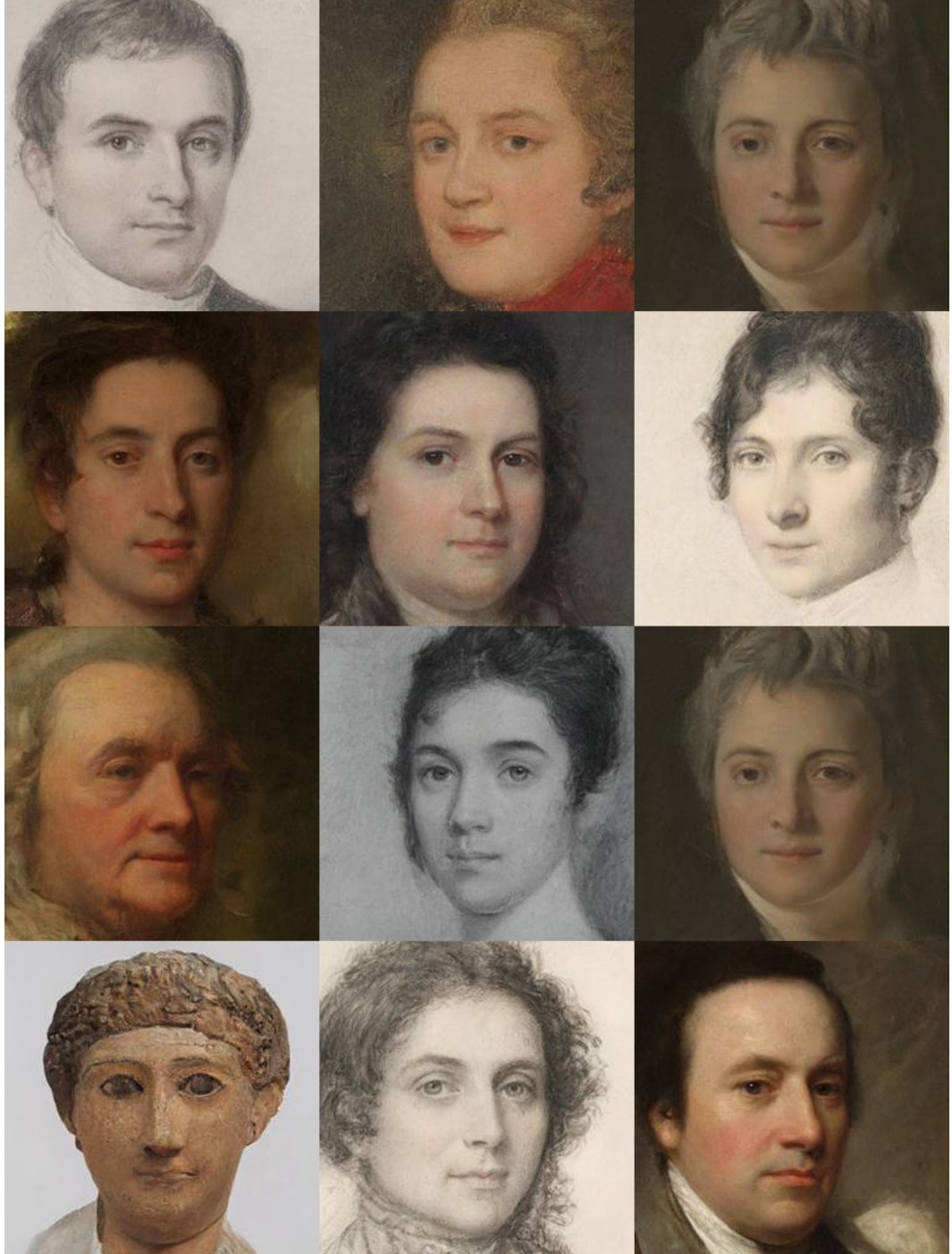

Figure 2: Generated Images on MetFace

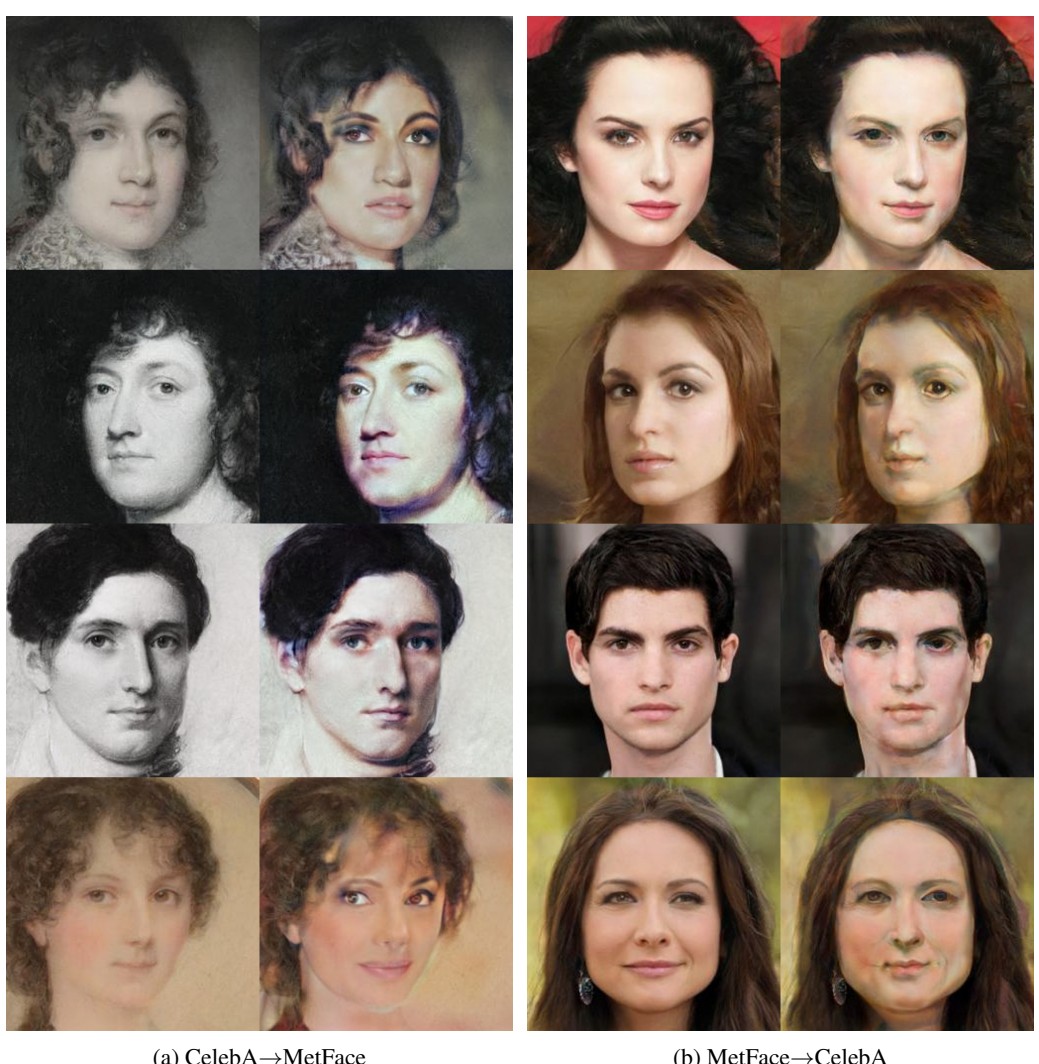

(a) CelebA→MetFace                     (b) MetFace→CelebA

Figure 3: (a) The image translation from CelebA to MetFace, via solving $dx = [f_\theta(x(t), t, c = 1) - f_\theta(x(t), t, c = 0)]dt$. (b) The reverse translation from MetFace to CelebA.

## 2.2    $256 \times 256$ Images Generated by a Conditional ScoreFlow Trained on AFHQ

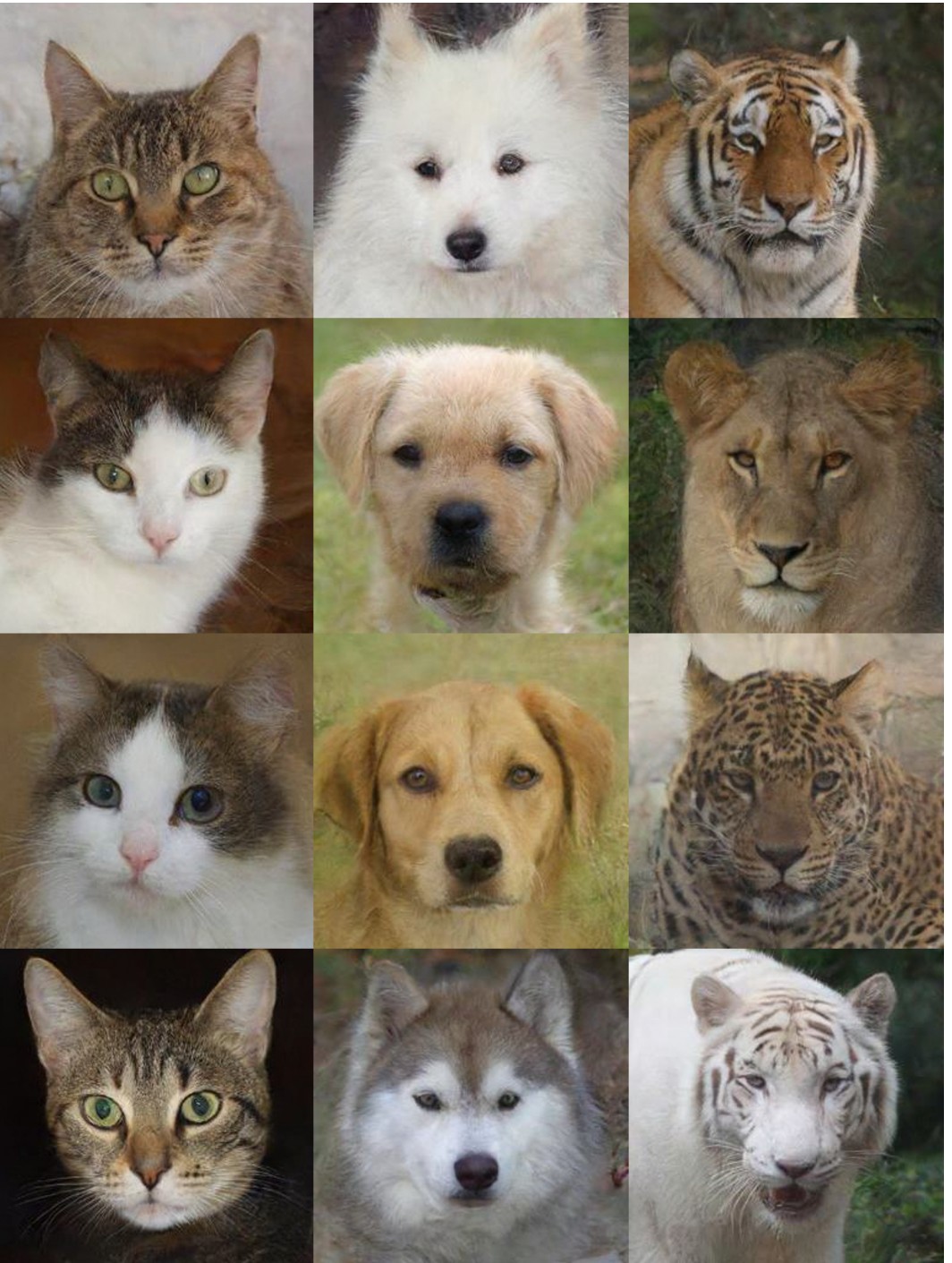

Figure 4: Generated Images on AFHQ