# OpenReview forum: "ScoreFlow: Bridging Score and Neural ODE for Reversible Generative Modeling"
_ICLR.cc/2024/Conference — ICLR 2024 Conference Withdrawn Submission_

### Official Review · Reviewer_y8UH · 2023-10-24

**Soundness:** 3 good
**Presentation:** 1 poor
**Contribution:** 1 poor
**Rating:** 3
**Confidence:** 4

**Summary:**

This paper...
- proposes ScoreFlow, a Neural ODE model for generating data from noise,
- proposes a path-constrained loss for reducing truncation error,
- proposes a method for image-to-image translation by subtracting two ODE vector fields.

Experiments results show that...
- ScoreFlow achieves comparable performance in unconditional generation on CIFAR10,
- ScoreFlow is able to translate between e.g., Cat & Wild and CelebA & MetFace.

**Strengths:**

- ScoreFlow is able to perform unconditional generation as well as image-to-image translation.

**Weaknesses:**

**Weakness 1 : Lack of Theoretical Novelty**

- Lemma 3.1 and the first statement of Theorem 3.1 simply re-states the relation between Variance Exploding (VE) SDE and probability flow (PF) ODE in [1]. For instance, observe that the perturbation kernel (14) in this paper is identical to the perturbation kernel of VESDE (9) in [1]. **Consequently, unconditional ScoreFlow from noise to data is identical to PF ODE of VESDE.**
- Image-to-image translation by concatenating two neural ODEs has already been explored in [2]. While ScoreFlow and [2] are subtly different in the aspect that ScoreFlow uses difference of scores while [2] uses scores individually, they both ultimately translate between two images which share the same "latent noise". **Hence, I don't see any practical or theoretical benefit of using ScoreFlow over DDIB.**
- One-step approximation of the path-constrained loss in this paper is just score matching with a particular choice of weighing coefficient $\lambda(t)$. Consider the following calculation, where in the first line, we use the "one-step approximation" proposed by the authors:

$\mathbb{E}\_{t,x(t)} [\lambda(t) \|\| x(t) - \bar{x}(t) \|\|^2 ] \approx \mathbb{E}\_{t,x(t)} [\lambda(t) \|\| (x_1 + t f_\theta(\bar{x}(t),t)) - \bar{x}(t) \|\|^2 ]$

$= \mathbb{E}\_{t,x(t)} [\lambda(t) \|\| (x_1 + t f_\theta(\bar{x}(t),t)) - ((1-t) x_1 + t x_2) \|\|^2 ]$

$= \mathbb{E}\_{t,x(t)} [\lambda(t) \|\| t f_\theta(\bar{x}(t),t) - t (x_2 - x_1) \|\|^2 ]$

$= \mathbb{E}\_{t,x(t)} [\lambda(t) t^2 \|\| f_\theta(\bar{x}(t),t) - (x_2 - x_1) \|\|^2 ]$

[1] Score-Based Generative Modeling through Stochastic Differential Equations, Song et al., ICLR, 2021.

[2] Dual Diffusion Implicit Bridges for Image-to-Image Translation, Su et al., ICLR, 2023.

**Weakness 2 : Weak Experiment Results**
- Given that ScoreFlow is equivalent to previous score-based models, I don't see any meaning in the experiment results. For instance, unconditional ScoreFlow from noise to data in Table 1 is simply a re-implementation of PF ODE for VESDE.
- Likewise, image-translation results must be theoretically identical to DDIB translation results.

**Questions:**

- What is the reason for introducing $\sigma_t$ in Theorem 3.1 if we ultimately use (18), which is identical to Rectified Flow?

---

### Official Review · Reviewer_7nbi · 2023-10-30

**Soundness:** 1 poor
**Presentation:** 2 fair
**Contribution:** 1 poor
**Rating:** 3
**Confidence:** 4

**Summary:**

This paper proposes ScoreFlow, which learns generative models from a ODE family whose velocity is defined as the devision of two probability density functions of the two distributions. Although the authors shows intriguing empirical results, the methods proposed in the paper are equivalent (at least in a trivial way) to previous works.

**Strengths:**

1. The empirical performance of ScoreFlow on CIFAR 10 is very competitive to state-of-the-arts. It demonstrates the effectiveness of ScoreFlow.
2. The generated images are clear and beautiful, with intricate details, showing the feasibility of ScoreFlow in generative modeling.

**Weaknesses:**

1. The math notations are confusing, leading to possibly wrong derivations. Most of the potential mistakes are around Eq. (11) to Eq. (14) in the Appendix. For example, in Eq. (12), for two random samples from $\pi_1$ and $\pi_2$ respectively,  their corresponding latent noise $\sigma_t \epsilon$ are not necessarily the same, therefore this noise term cannot be cancelled in Eq. (13). Moreover, in Eq. (13), despite that the notations inside the integral are both $x(t)$, the paths of the two ODEs are fundamentally different ( for the same $t$, $x(t)$ is different), and the two integrals cannot be merged together.


2. Path constraint loss is equivalent to weighted Rectified Flow loss (see derivation below), which has already been discussed in Eq. (6) of [1]. Although I appreciate the empirical results that weighing the loss by $\lambda(t) = \frac{1}{t}$ (that equals $w_t=t$) improves the FID and IS on CIFAR10, I believe more systematic investigation on the weighing coefficients is required to make solid contribution to the field.

    *Derivation of Loss (19):*  $$ x(t) = x_1 + t f(\bar{x}(t), t), \bar{x}(t) = t x_2 + (1-t) x_1  \Rightarrow E[\lambda(t) || x(t) - \bar{x}(t) ||^2] = E[\lambda(t) || x_1 + t f(\bar{x}(t), t) - t x_2 - (1-t) x_1 ||^2] = E[\lambda(t) t^2 || f(\bar{x}(t), t) - (x_2 - x_1) ||^2]$$


3. The image-to-image translation algorithm boils down to generate images with two generative models using the same latent noise, since the pipeline starts from finding the noise $z_0$ of $x^{c_1}$ with the first generative model $f(x_t, t, c_1)$ and then seeks the corresponding $x^{c_2}$ by simulating the second generative model $f(x_t, t, c_2)$. This has been proposed and discussed by multiple previous works, e.g., [2].


Overall, I think the mathematical correctness is doubtful and the novelty of the proposed methods is limited, making it apparently below the bar of ICLR.

[1] Rectified Flow (arXiv): https://arxiv.org/pdf/2209.03003.pdf

[2] UNDERSTANDING DDPM LATENT CODES THROUGH OPTIMAL TRANSPORT: https://openreview.net/pdf?id=6PIrhAx1j4i

**Questions:**

Please refer to Weakness.

---

### Official Review · Reviewer_U1NS · 2023-10-31

**Soundness:** 3 good
**Presentation:** 3 good
**Contribution:** 2 fair
**Rating:** 5
**Confidence:** 4

**Summary:**

This paper proposes a "ScoreFlow" method to training Neural ODEs. In particular, the paper models the difference in score between a $p_1$ and $p_2$ which allows one to translate between the two distributions. In particular, one can train a network for mapping between $b \to p_1$ and $b \to p_2$ and then combine these methods.

**Strengths:**

* The paper very theoretically clean.
* The paper draws an interesting connection between the score function of an unconstrained generation and distribution translation, which allows it to motivate and generalize previous work.
* By doing this new interpretation, the paper opens up new possibilities like classifier free guidance (as introduced in the paper)

**Weaknesses:**

* The experiments are somewhat incomplete. In particular, while numerical results are shown for CIFAR-10, several baselines are conspicuously missing like even the VESDE and more recent baselines (EDM, etc...). Additionally, the translation examples are qualitative (and as such can be cherry-picked); the paper should include some image translation exapmles.
* Although classifier free guidance is mentioned, it is not tested.
* What is the relationship between this and Doob's H transform? Doob's H transform also allows one to translate from one distribution to another with a variant of the score function, so I suspect that this might be the unifying principle. Note that rectified flow has a score function/diffusion interpretation (see [1]).

[1] https://arxiv.org/abs/2303.00848

**Questions:**

Please consider changing the name of "ScoreFlow", as this already exists as a name in a well-known paper [1].

[1] https://github.com/yang-song/score_flow

---

### Official Review · Reviewer_Tmwk · 2023-11-01

**Soundness:** 3 good
**Presentation:** 3 good
**Contribution:** 2 fair
**Rating:** 5
**Confidence:** 4

**Summary:**

This paper presents a generative modeling framework based on Neural ODE. The main idea is to guide the drift function in neural ODE with the score function of the data distribution. This greatly simplifies the underlying dynamics of the neural ODE system, giving analytic transition densities and thus leading to a closed-form drift function target to be approximated with a neural network. Moreover, thanks to the nice linear interpolation form of the drift, one can easily extend the proposed model to noise-to-data generation, data-to-data connection, or conditional generations. The proposed model, named as scoreflow, is demonstrated on some image generation benchmarks.

**Strengths:**

- The paper is well-written and easy to follow.
- The proposed model is simple and easy to implement.
- It is good to have a single model capable of solving various generation tasks.

**Weaknesses:**

- The contribution of the paper is rather obscure. What is proposed as a "ScoreFlow" (equation (7)) is a minor modification (to generalize the noise part to arbitrary distribution) of probability flow ODE (Song et al., 2019), with the parameterization suggested in (Karras et al., 2022), although the paper describes as if they have come up with this from the standpoint of improving neural ODE. Even further, as described at the bottom of page 5, the coefficient $\dot{\sigma}_t/\sigma_t$ is fixed as a constant, making it identical to RectifiedFlow (Liu et al., 2023). Considering the usual observation on the importance of variance scheduling (e.g., Karras et al., 2022), this might potentially make the model less flexible, so this choice is quite disappointing.

- Limited experiments. The unconditional image generation was done only for CIFAR10, which is relatively low-resolution and easy data to compare score-based generative models. Considering also the relationship between ScoreFlow and PF-ODE proposed in Karras et al., (i.e., EDM), it would be worth trying to compare those to methods (EDM uses a second-order solver so this should be accounted).

- No quantitative evaluation for image translation & interpolation experiments. While the model can indeed be adapted for those tasks easily, the actual performance is a different thing. Showing some samples does not tell anything about the overall quality, and moreover there is no comparison to existing methods.

- I like the fact that a single class-conditional model can be used to define an image translation model between arbitrary classes without further adjustment (section 3.3). But I fail to see corresponding demonstration. It would be good to compare the single class-conditional model adapted in that way to an image translation model specifically trained for individual class combinations.

**Questions:**

- In the path-constrained loss, the ODE solution $x(t)$ is approximated with a one-step approximation. While this would be accurate when the learned drift $f_\theta(x(t), t)$ is indeed close to the true drift, for instance at the beginning of the training, this approximation may be severely wrong when $t$ is close to one. Any explanation for this?
- The path-constrained loss is described to be a distinction from RectifiedFlow. Does it actually help? If so, how good is it compared to the vanilla training loss (equation (18))?
- Can you also apply reflow for ScoreFlow to further improve the sampling efficiency?
- Minor point, the name "ScoreFlow" has already been taken by the paper, Song et al., Maximum Likelihood Training of Score-Based Diffusion Models, 2021. I suggest to consider a different name to avoid confusion in the literature.

---

### Official Review · Reviewer_VG7T · 2023-11-02

**Soundness:** 2 fair
**Presentation:** 2 fair
**Contribution:** 1 poor
**Rating:** 1
**Confidence:** 4

**Summary:**

The authors claim that they propose a novel training method for neural ODE-based generative models, whose training is similar to either diffusion-based models or rectified flows.


To do that, the paper first highlights the probability flow formulation of the time-reversed stochastic differential equations (time-reversed SDEs) in the continuous-time diffusion-based generative models. Furthermore, the paper emphasizes that the probability flow’s drift term (i.e., vector field term) corresponds to the optimal solution of the generative ODE for the given forward (linear) diffusion. Thus, the paper proposes directly minimizing the mean squared error between the model and the probability flow’s drift term. Note that the analytical form of the probability flow’s drift term can be obtained when the forward diffusion is linear, i.e., $p(x_t | x_0)$ is a normal distribution for all $t$ in $[0, T]$.

The paper claims that, unlike conventional maximum likelihood methods, the proposed method is computationally efficient and thus scales well since the proposed method doesn’t require likelihood estimation.

The paper also proposes a regularization method to accelerate the model's training, called ‘path-constrained loss’.

The paper also proposes various application scenarios of the proposed method, such as class image-to-image translations.

Finally, the paper demonstrates the performance of the proposed methods in various benchmark datasets.

**Strengths:**

N/A

**Weaknesses:**

In general, I find that the paper is not qualified for the ICLR conference. Unfortunately, the paper introduces a method very similar to flow matching methods, including Y. Lipman et al. '22, flow matching for generative modeling, and other follow-up works. For example, the Theorem 3.1. in the paper corresponds to the Theorem 3 in Y. Lipman et al. '22. Most importantly, the paper didn't provide any novelty (or connections) compared to the previous literature. Other application scenarios introduced in the submission can be found in the previous works.

On the other hand, the path-constrained loss can be treated as a novelty compared to the aforementioned flow-matching families. However, such discussions have also been well discussed in some of previous works, including C.-H. Lai et al. '23.



Y. Lipman et al. '22, Flow Matching for Generative Modeling
C.-H. Lai et al. '23, On the Equivalence of Consistency-Type Models: Consistency Models, Consistent Diffusion Models, and Fokker-Planck Regularization

**Questions:**

N/A